



# Evaluating the suitability of the consumer low-cost Parrot Flower Power soil moisture sensor for scientific environmental applications

Angelika Xaver[1], Luca Zappa[1], Gerhard Rab[2], Isabella Pfeil[1,2], Mariette Vreugdenhil[1,2], Drew Hemment[3], and Wouter A. Dorigo[1]

[1]Department of Geodesy and Geoinformation, TU Wien, Gußhausstraße 27-29/E120, 1040 Vienna, Austria
[2]Centre of Water Resource Systems, TU Wien, 1040 Vienna, Austria
[3]Edinburgh Futures Institute and Edinburgh College of Art, University of Edinburgh, Edinburgh, United Kingdom

**Correspondence:** Angelika Xaver, Wouter A. Dorigo (Angelika.Xaver@geo.tuwien.ac.at, Wouter.Dorigo@geo.tuwien.ac.at)

**Abstract.** Citizen science, scientific work and data collection conducted by or with non-experts, is rapidly growing. Although the potential of citizen science activities to generate enormous amounts of data otherwise not feasible is widely recognized, the obtained data are often treated with caution and skepticism. Their quality and reliability is not fully trusted since they are obtained by non-experts using low-cost instruments or scientifically non-verified methods. In this study, we evaluate the perfor-

mance of Parrot's Flower Power soil moisture sensor used within the European citizen science project, the GROW Observatory (GROW; https://growobservatory.org). The aim of GROW is to enable scientists to validate satellite-based soil moisture products at an unprecedented high spatial resolution through crowdsourced data. To this end, it has mobilized thousands of citizens across Europe in science and climate actions, including hundreds who have been empowered to monitor soil moisture and other environmental variables within twenty four high-density clusters around Europe covering different climate and soil conditions.

Clearly, to serve as reference dataset, the quality of ground observations is crucial, especially if obtained from low-cost sensors. To investigate the accuracy of such measurements, the Flower Power sensors were evaluated in the lab and field. For the field trials, they were installed alongside professional soil moisture probes in the Hydrological Open Air Laboratory (HOAL) in Petzenkirchen, Austria. We assessed the skill of the low cost sensors against the professional probes using various methods. Apart from common statistical metrics like correlation, bias and root-mean-square difference, we investigated and compared

the temporal stability, soil moisture memory, and the flagging statistics based on the International Soil Moisture Network (ISMN) quality indicators. We found a low inter-sensor variation in the lab and a high temporal agreement with the professional sensors in the field. The results of soil moisture memory and the ISMN quality flags analysis are in a comparable range for the low-cost and professional probes, only the temporal stability analysis shows a contrasting outcome. We demonstrate that low-cost sensors can be used to generate a dataset valuable for environmental monitoring and satellite validation and thus

provide the basis for citizen-based soil moisture science.

## 1   Introduction

The importance of soil moisture for the hydrological cycle and the climate system is well-known (Seneviratne et al., 2010). For decades techniques and instruments to measure and monitor water content in the soil at different spatial and temporal





scales have been developed and improved. One of the most promising means to observe the state of soil moisture in the top

soil layer on a global and long-term scale is microwave remote sensing. A range of active and passive satellite instruments have collected and are still gathering data, forming the basis for various soil moisture products including, but not limited to, Advanced Microwace Scanning Radiometer (AMRS-E) Land Parameter Retrieval Model (LPRM), Level 2 Passive Soil Moisture Product (L2SMP), Advanced SCATterometer (ASCAT) Soil Moisture Product, Sentinel-1 Surface Soil Moisture, and ESA Climate Change Initiative (CCI) Soil Moisture, (e.g., de R.A.M. Jeu et al., 2008; Chan et al., 2016; Wagner et al.,

2013; Bauer-Marschallinger et al., 2019; Dorigo et al., 2017). Ground observations play a crucial role in evaluating and further improving these satellite products, particularly with respect to novel satellite missions that provide data at unprecedented spatial resolution, e.g. Copernicus Sentinel-1. Although several in situ soil moisture monitoring stations and networks have been established around the globe, ranging from short-term campaigns to long-term observations, and many of them provide their valuable data through the International Soil Moisture Network (ISMN; Dorigo et al., 2011), acquisition, installation, and

maintenance of the professional equipment is costly and laborious. Thus, the coverage of such stations is mostly limited in space, density, and duration.

A large number of measurement techniques for estimating soil moisture at the point scale is available Ochsner et al. (2013). Most common are methods based on electromagnetic waves, including time domain reflectometry (TDR) (e.g., Robinson et al., 2008) and capacitance sensors (e.g., Bogena et al., 2007). While TDR sensors are known for their higher accuracy, the advan-

tage of capacitance probes is that they are less expensive while still providing sufficiently reliable readings (e.g., Kojima et al., 2016). For this reason, capacitance sensors have gained popularity and have often been investigated. While commonly being referred to as "low-cost" sensors (e.g., Mittelbach et al., 2011; Bogena et al., 2007; Kizito et al., 2008; Domínguez-Niño et al., 2019; González-Teruel et al., 2019; Matula et al., 2016), deploying them in high numbers and densities is still too expensive. The alternative is either to design and develop a completely new sensor (e.g., González-Teruel et al., 2019; Kojima et al., 2016),

which is a time consuming and challenging task, or to make use of existing commercial products, which are not designed for scientific use and are in fact low-cost sensors.

The latter idea was pursued by the European citizen science project GROW Observatory (GROW; https://growobservatory.org/). The aim of GROW is to generate a vast in situ soil moisture network across Europe maintained by citizens using the commercial soil moisture sensor Flower Power (FP), produced by the French company Parrot SA, in order to support remote sensing

scientists. The involvement of non-experts in collecting scientific data has a long history, especially in the field of biology, astronomy and meteorology (Silvertown, 2009). In recent years, citizen science has aimed to go beyond mere data collection to introduce non-scientists to the research design, data interpretation and application of results, which is also the scope of the GROW Observatory.

In this study, we assess the performance of the low-cost soil moisture sensor Flower Power (FP) from Parrot SA, used

within the GROW Observatory. In particular, we evaluate the FP sensor under laboratory conditions and in an extensive field campaign performed in an agricultural catchment in Austria. The main objectives are to (i) assess the agreement of FP readings with gravimetric measurements, (ii) determine the inter-sensor variability, (iii) test the sensors' suitability for outdoor usage,





(iv) assess their agreement with professional soil moisture probes, and finally (v) conclude if the FP sensors are suitable for scientific environmental applications, in particular satellite validation.

## 2   Data

### 2.1   Low-cost sensor

The low-cost sensor used in this study is the Flower Power (FP) sensor (Parrot, 2019c), a consumer product of the French company Parrot SA. It was designed for home use, indoor and outdoor, with the purpose of observing the condition of the user's plants. The sensor provides information about soil humidity, air temperature, light intensity, and fertilizer content in the soil (Fig. 1). The soil water content is measured with a capacitance probe, consisting of two flat rods of ten centimeters length.

**Table 1.** Variables, unit, range, and accuracy of the used soil moisture sensors as specified by the manufacturer.

| Producer | Sensor name | Variable | Unit | Range | Accuracy |
|---|---|---|---|---|---|
| Parrot SA | Flower Power | Air Temperature | °C | -5 to +50 °C | $\pm$1.5 °C |
| | | | (°F) | (23 to 131 °F) | ($\pm$2.7 °F) |
| | | Fertilizer Level | mS cm$^{-1}$ | 0 to 10 mS cm$^{-1}$ | $\pm$20% |
| | | Light Intensity | mol m$^{-2}$d$^{-1}$ | 0.13 to 104 mol m$^{-2}$d$^{-1}$ | $\pm$15% |
| | | Soil Moisture | vol% | 0 to 50 vol% | $\pm$3% |
| sceme.de GmbH | TDT SPADE | Dielectric Constant | - | 1 to 85 | $\pm$4% |
| | | Soil Temperature | °C | -10 to +85 °C | $\pm$0.5 °C |
| METER Group, Inc. USA | ECH2O 5TM | Soil Moisture | m$^3$m$^{-3}$ | 0.0 to 1.0 m$^3$m$^{-3}$ | $\pm$0.03m$^3$m$^{-3}$ |
| | | Soil Temperature | °C | -40 to +60 °C | $\pm$0.1 °C |

This method, which is already known for many years (Dean et al., 1987), uses an oscillator to propagate an electromagnetic signal through the rods into the soil. The charging time of the electromagnetic field is related to the capacitance of the soil, which in turn is related to its dielectric permittivity. Consequently, as the dielectric permittivity is sensitive to water, the soil moisture content can be estimated. Kizito et al. (2008) summarized that the measurement frequency of capacitance probes is one of the main factors influencing the sensitivity of their measurements to soil texture, electrical conductivity, and temperature. Unfortunately, no information about the measurement frequency of the FP sensor is supplied by the manufacturer.

Two metal domes sitting on top of the FP sensor's prongs estimate the fertilizer level by measuring the electric conductivity. When vertically installed in soil as designated, the prongs are completely sunk into the ground and only a plastic fork of an approximate height of nine centimeters with two ends remains above the soil. One end holds the battery compartment for one AAA battery 1.5V, the other end an air temperature and light sensor, measuring visible light in the wavelength range between 400 and 700 nm. Observations of all four variables are automatically taken every 15 minutes and stored on the device, where data can be saved up to 80 days before they get overwritten. One battery provides the sensor with enough energy to continue



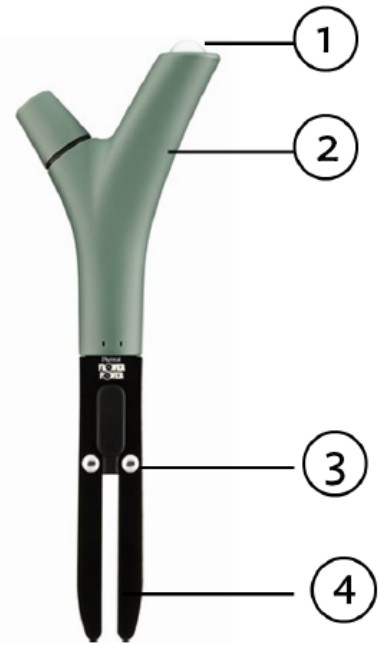

**Figure 1.** Schematic of the FP sensor: 1. light intensity, 2. ambient temperature, 3. fertilizer level, and 4. soil moisture sensor (Parrot, 2019d).

measuring for six to twelve months, depending on the weather conditions. Details about the available variables, their units, range, and accuracy indicated by the manufacturer can be found in Table 1.

80    The observations collected by the FP sensor can only be accessed by the Parrot Flower Power app, which is available for iOS and Android smartphones. When positioned in close vicinity to the sensor the app automatically connects via the Bluetooth Low Energy (LE) protocol and collects the stored data from the sensor. As soon as an internet connection is available the observations are uploaded to the Parrot Cloud (Parrot, 2019a). After this process the collected data can be viewed in the time series visualization of the app, in addition to the Live mode, where only current values are visible. From the Parrot Cloud the
85    data can be obtained in .csv format through an API provided by Parrot SA (Parrot, 2019b).

The Parrot Flower Power app and cloud services were integrated as part of the GROW ecosystem of services. GROW furthermore augmented the FP sensor with social and scientific services to address the needs of citzens, scientists and policy makers participating in the citizens' observatory. That wider system architecture is outside the scope of this paper.

## 2.2    Reference sensors

90    Two conventional sensors were used to evaluate soil water content measurements of the FP sensor in the field: the Time Domain Transmissivity (TDT) SPADE sensor (Qu et al., 2013; sceme.de GmbH, 2019) and the 5TM (ECH2O 5TM from METER Group, Inc. USA) based on capacitance technology. The SPADE sensor consists of a sensor head of 8 cm length and



a flat prong 3 cm wide and 12 cm long containing the closed transmission line of the ring oscillator. The frequency of the ring oscillator depends on the dielectric constant of the surrounding material, i.e. the soil. The determined dielectric constant can range from 1 to 85 with a relative accuracy given as $\pm$ 4% by the manufacturer (sceme.de GmbH, 2019). It also measures soil temperature within a range from -10 to +85 °C, with an accuracy of $\pm$ 0.5 °C (Qu et al., 2013). Observations are transmitted and stored according to the SoilNet wireless network technology developed at Forschungszentrum Jülich (Bogena et al., 2010). The 5TM probe consists of three flat rods and has a total length of 10.9 cm and width of 3.4 cm. The mineral soil calibration, supplied by the manufacturer, is used, providing soil water content within a range of 0.0 to 1.0 $m^3m^{-3}$, a resolution of 0.0008 $m^3m^{-3}$, and an accuracy of $\pm$ 0.03 $m^3m^{-3}$, specified by the producer (METER Group, 2019). Soil temperature measured by the 5TM probe can range from -40 to +60 °C and is given with a resolution of 0.1 °C and an accuracy of $\pm1$ °C. Data are logged with the Em50G Data Logger (METER Group, Inc. USA).

Ambient temperature readings of the low-cost sensor are compared with the conventional Vaisala HUMICAP humidity and temperature probe HMP155 (Vaisala Oyj, Finland), which has a measurement range from -80 to +60 °C (Vaisala, 2019). The CNR4 Net-Radiometer (Kipp & Zonen B.V., Netherlands), a four component net radiometer, serves as reference for the FP sensor light readings. In particular in this study, we use the incoming shortwave radiation which is observed within a spectral range from 300 to 2800 nm, with a sensitivity of 10 to 20 $\mu$V $W^{-1}m^{-2}$ (Kipp & Zonen, 2019). The observed spectral range differs significantly from that of the FP sensor, thus the observations are not comparable in absolute terms. Nevertheless, the relative temporal agreement between the FP sensor and the CNR4 can be determined.

## 2.3 Study Area and sensor set-up

The study area is the Hydrological Open Air Laboratory (HOAL, http://hoal.hydrology.at; Blöschl et al., 2016) located in Petzenkirchen (48°9' N, 15°9' E), Austria. The gentle hills of the agricultural catchment, covering an area of 66 ha, are situated at an average elevation of 296 m above sea level. The climate is characterized by a mean annual temperature of 9.5 °C and a mean annual precipitation of 823 mm per year (Blöschl et al., 2016). Blöschl et al. describe the different monitoring stations and devices installed in the HOAL catchment in order to investigate and answer a variety of scientific hypotheses. While four rain gauges were already set up in 2010, the weather station, located in the centre of the catchment, was fully equipped with instruments in 2012. Among other devices the weather station accommodates the air temperature probe HMP155 and the four component net radiometer CNR4, installed at a height of 2 m and 2.5 m respectively. Since 2013, the catchment is equipped with 20 permanent and 11 temporary soil moisture stations (Fig. 2; Vreugdenhil et al., 2013). The permanent stations are located in pasture and forest, the temporary stations are installed in agricultural fields and have to be removed and re-installed on an irregular basis to allow for field management. All except one of the soil moisture stations are equipped with the SPADE TDT sensors. One station, namely "Hoal_D3", located next to the weather station, uses the 5TM sensor to observe soil water content and soil temperature in 0.05 and 0.10 m depth and serves as a reference for the observations of the SPADE TDT sensors. Except for this station, all soil moisture monitoring stations are equipped with horizontally installed sensors at four different depths: 0.05 m, 0.10 m, 0.20 m, and 0.50 m. In addition, two stations have a sensor in 1.00 m depth available. For

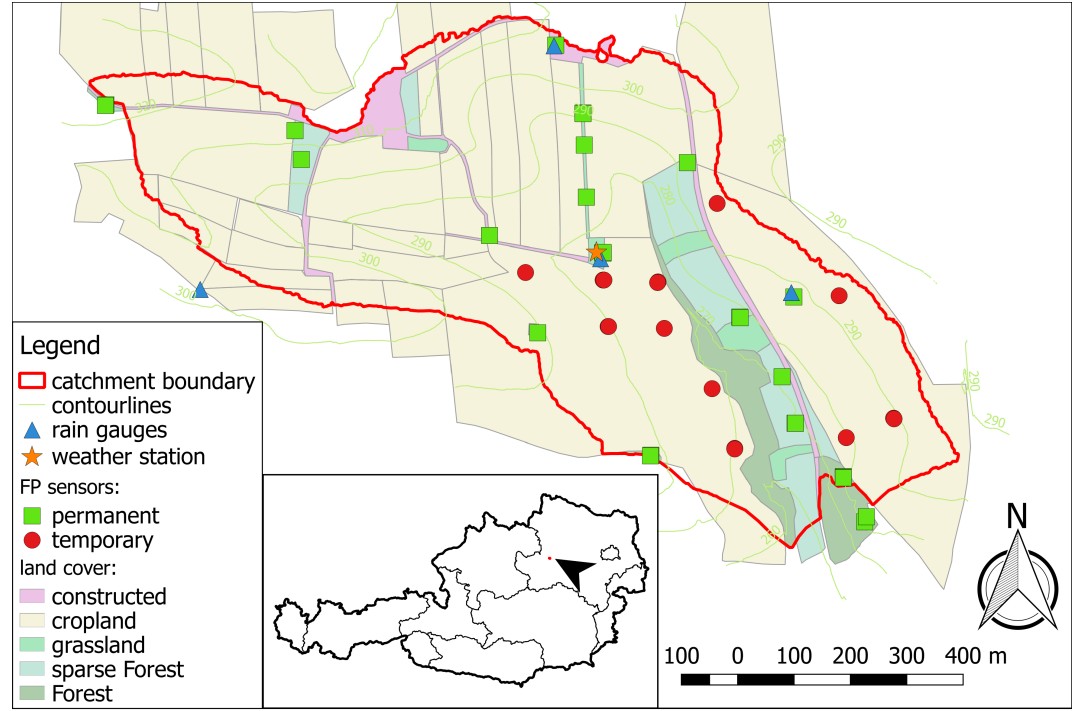

**Figure 2.** Locations of the permanent and temporary FP sensors in the HOAL catchment, marked by green squares and red circles, respectively.

nine stations, two probes collect soil moisture readings simultaneously in a depth of 0.05 m and eleven stations are equipped with two probes in a depth of 0.10 m.

In Spring 2017, 37 FP sensors were placed alongside the 31 professional sensors in the HOAL catchment (Fig. 2), allowing for cross-validation between FP sensors at six locations. Between spring 2017 and spring 2019, 15 additional sensors were installed, seven to replace sensors with a failure and eight to allow for further cross-validation between FP sensors. Thus, in total 52 FP sensors were available to evaluate their performance with respect to 31 professional probes. 50 sensors were installed as designated in a vertical position, providing information about the water content of the upper ten centimeters of soil. Two sensors were buried in a depth of 0.05 m in a horizontal position to allow for a more direct comparison with the horizontally positioned professional probes. The exact horizontal distance between the FP sensors and the professional probes is unknown due to the different set up times. The position of the professional probes could only be estimated based on the wires and data logging devices above the ground. Thus, the maximum possible distance between both sensors is estimated to be one meter, but is assumed to be much less in most cases. In May 2018, one FP sensor (TUW90a) was attached to a rod in 1.7 m height next to the weather station with the purpose to collect light intensity.

The observation period of the FP sensors varies due to different installation dates and temporary removal because of field



management, and ranges from a few months to 1.5 years. Three FP sensors were excluded from the analysis due to sensor malfunction and too short observation periods. For similar reasons six professional sensors at the depth of 0.05 m and eleven sensors at the depth of 0.10 m were neglected during the analysis. In general, only sensor pairs with more than three months of simultaneous observations were not considered for further analysis. All investigations were performed on hourly values. For convenience all soil moisture values and results are expressed in $m^3m^{-3}$.

## 3    Methods

Two different approaches were applied to evaluate the performance of Parrot's Flower Power sensors. First, observations of five FP sensors were compared to gravimetrically measured soil water content in the laboratory (Calibration, see Sect. 3.1). Second, the Parrot FP sensors were installed in the field and their measurements were compared to those from professional soil moisture probes by using four different metrics: 1) conventional statistical measures, 2) relative mean difference, 3) auto-correlation,

and 4) independent quality indicators (Sect. 3.2). All comparisons were performed on hourly observations.

### 3.1    Calibration

The customer of the FP sensor has to use the predefined calibration for potting soil, as this sensor is a commercial product designed to monitor potted plants, and cannot choose between different calibration curves depending on local soil conditions which are typically available for professional sensors. Thus, in summer 2018, five Parrot FP sensors were calibrated for

the dominant soil texture class present in the HOAL catchment under laboratory conditions, following a calibration procedure developed at the Institute for Land and Water Management Research (Federal Agency for Water Management Austria, Petzenkirchen, Austria). A homogeneous soil sample from the catchment with particle sizes of less than 2 mm was used for this exercise. The soil sample, characterized as silty clay loam, according to the United States Department of Agriculture (USDA) standards (Soil Science Division Staff, 2017), consists of 27.5 wt% clay, 65 wt% silt, and 7.5 wt% sand. The air-dry sample

was divided into five equal parts in order to investigate sensor performance for five different amounts of soil water content: air-dry, 0.08 $m^3m^{-3}$, 0.20 $m^3m^{-3}$, 0.30 $m^3m^{-3}$, and fully saturated. The respective amount of water was added to the air-dry soil samples, manually mixed, and left to rest under protected conditions over night to ensure a homogeneous water content throughout the entire sample. A Boyle-Mariotte bottle was used to obtain the fully saturated soil sample. The gravimetric water content of the five soil samples was determined in order to conclude on the respective soil volume needed to fill five cylinders

(15 cm diameter, 30 cm height) with a bulk density of 1.3 g cm$^{-3}$. The five cylinders were filled with the five soil samples following the same technique: One fifth of the soil sample was added to the cylinder and manually compacted with a proctor plate until the soil specimen reached a height of 5 cm in the cylinder and thus the desired density. The soil surface was roughened and the same procedure was repeated for the remaining four fifth until the cylinder was filled up to a height of 25 cm. After all five cylinders were filled, the five FP sensors were subsequently installed in undisturbed spots of the five soil samples.

The according time stamp and soil moisture readings in Live mode of the Parrot App were recorded immediately and again before the FP sensor was removed. The actual soil moisture readings were extracted afterwards from the Parrot cloud. The FP





sensors were left in each soil sample for at least 20 minutes to ensure that at the minimum two data records per soil sample were automatically stored on the sensor. As soon as the sensor measurements were finished small soil samples were extracted from the five cylinders and their water content was derived gravimetrically to serve as reference for the sensor readings.

### 175 3.2 Sensor-to-sensor Comparison

The measurements collected by the FP sensors in the field were compared to the sensor readings taken with professional probes described in Sect. 2.2.

#### 3.2.1 Temporal agreement

Commonly known statistical measures including Pearson correlation coefficient (R), standard deviation, bias, root-mean-
squared difference (RMSD), and unbiased root-mean-squared difference (uRMSD) were used to determine the agreement between the in situ soil moisture recordings of the FP and the professional sensors. Observations from the SPADE and 5TM sensors at 5 and 10 cm depth served as a reference. 33 sensor pairs with an average observation period of ten months were used for comparison between FP sensors and professional probes at 0.05 m depth (Table A1). For the comparison with the professional sensors at 0.10 m depth 27 sensor pairs were analyzed.
In addition to the soil water content, the air temperature measurements of five FP sensors were evaluated with respect to the HMP155 air temperature and relative humidity sensor positioned in a height of 2 m at the weather station of the HOAL catchment. Four of the five FP sensors investigated were conventionally installed in soil and encompass all four land cover classes of the catchment (grassland, cropland, sparse forest, and forest). The fifth sensor is located at the weather station 1.7 m above the soil and was also used to evaluate the light intensity observations against the downwelling shortwave radiation obtained by
the CNR4 Net-Radiometer.

#### 3.2.2 Temporal stability

The concept of spatial soil moisture patterns being persistent over time is called temporal stability and was introduced by (Vachaud et al., 1985) and often investigated (e.g., Cosh, 2004; Brocca et al., 2010, 2012; Caldwell et al., 2018). Temporally stable and representative sensors for an entire area or catchment with respect to the average conditions in that area can be
identified through this method, even as locations that are consistently wetter or drier than the overall average. For this purpose the mean relative difference (MRD) was used, according to the following definition

$$MRD_i = \frac{1}{t} \sum_{j=1}^{t} \frac{\theta_{i,j} - \bar{\theta}_j}{\bar{\theta}_j} \qquad (1)$$

where $\theta_{i,j}$ is the water content at hour $j$ and sensor $i$, and $\bar{\theta}_j$ refers to the average over all sensors at hour $j$. Consequently, a negative (positive) $MRD_i$ means that sensor $i$ represents a drier (wetter) location with respect to the catchment average. The





sensor with the lowest absolute $MRD$ is the best estimate for the catchment average. The according variance $\sigma_i^2$ results from
the equation

$$\sigma_i^2 = \frac{1}{1-t}\sum_{j=1}^{t}\left(\frac{\theta_{i,j}-\bar{\bar{\theta}}_j}{\bar{\bar{\theta}}_j}-\bar{\theta}_i\right) \tag{2}$$

and measures the temporal stability of the relation between a sensor $i$ and the overall mean.

Our goal was to compare the soil moisture temporal stability of both sensor types, the low-cost and the professional sensors. In particular, we were interested whether equal sites were identified as dry and wet locations based on the both sensor sets. A total of 26 sensor pairs (low-cost and professional) formed the basis for the temporal stability analysis, only one sensor pair per location was used. In case of two suitable sensors of the same type (FP sensors or professional sensors) being available at the same location, the one with the longer observation period was chosen.

### 3.2.3 Soil Moisture Memory

The capability of soil to store information of an anomalous condition (e.g. rainfall, drought) long after its occurrence is commonly referred to as soil moisture memory (Delworth and Manabe, 1988). Memory timescales of root-zone soil moisture are typically in the order of a week to a few months, depending on soil properties and meteorological variables (Ghannam et al., 2016).

Soil moisture memory is commonly derived by the time-lagged auto-correlation given as

$$r(\tau) = e^{(-\tau/\lambda)} \tag{3}$$

where $\tau$ describes the time lag and $\lambda$ refers to the decay time scale or e-folding time, at which the auto-correlation $r$ reduces to $1/e$ (Delworth and Manabe, 1988). It represents a measure of how long a soil moisture anomaly can be detected and, more importantly, influences the atmosphere. For this reason, soil moisture memory is being well investigated and implemented by land-climate modelers (e.g., Maurer et al., 2001; Koster and Suarez, 2001; Sörensson and Berbery, 2015). The importance and potential of soil moisture memory is also recognized in other research areas. For example, Rebel et al. (2012) and Piles et al. (2018) used an auto-correlation analysis to compare the temporal dynamics captured by different soil moisture products, i.e. in situ, modeled, and satellite-based.

We applied this technique to the catchment average soil moisture derived from both sensor types, using the same amount of FP and professional sensors, and expect that any arising differences are mainly driven by the individual sensor characteristics since the temporal dynamics, i.e. anomalies, of the catchment are the same. Considering that less than two years of observations were available for the FP sensors, the soil moisture climatology and consequently the anomalies were derived based on moving averages of 35 days (Dorigo et al., 2015; Albergel et al., 2012). To ensure comparability the same method for deriving the climatology and anomalies was applied to the observations from the professional sensors.





### 3.2.4 Automated Quality Control

Dorigo et al. (2013) developed automated quality control procedures that were specifically designed for in situ soil moisture observations and which have been implemented in the International Soil Moisture Network (ISMN; https://ismn.geo.tuwien.ac.at/; Dorigo et al., 2011). The ISMN is an international initiative to make in situ soil moisture observations available in a harmonized format. The automated quality control procedures of the ISMN comprise thirteen different quality identifiers (C01–C03, D01–D10; see Table 3) based on three different approaches. First, a simple threshold based technique identifies values outside

a reasonable geophysical range (C01–C03). Second, a geophysical consistency test investigates the plausibility of soil moisture observations in connection with additional environmental variables such as soil temperature and precipitation (D01–D05). Third, suspicious measurements are detected based solely on the spectrum of the soil moisture time series (D06–D10). To every soil moisture observation one or more quality identifiers are attached. In case that none of the three approaches apply to the underlying observation, the identifier G (good) is assigned. For more details we refer to (Dorigo et al., 2013).

In this study, we applied the ISMN quality flags to both the data from the low-cost sensors and from the professional probes, with the purpose to compare the flagging statistics and investigate if the percentage of flagged observations differs substantially. In order to make the results comparable, the ISMN quality identifiers were applied to the same amount of FP and professional devices, i.e., one FP sensor per location.

## 4 Results and Discussions

### 4.1 Calibration results


  The gravimetrically determined and the observed soil water content by using the five FP sensors is shown in Fig. 3 and Table A2. For low soil water content from 0.03 m$^3$m$^{-3}$ (air-dry) to 0.20 m$^3$m$^{-3}$, the FP sensors consistently measure higher values than the actual water content. All five sensors show the highest (positive) deviation of 0.06 to 0.09 m$^3$m$^{-3}$ at the gravimetrically derived water content of 0.08 m$^3$m$^{-3}$. For the higher soil water content the deviation is much lower and ranges from 0.01 to 0.02 m$^3$m$^{-3}$.

The measurements of the five low-cost sensors are consistent, only sensor 196 is showing consistently higher values than the other four sensors. The overall inter-sensor variation is very low with only 0.01 m$^3$m$^{-3}$, which is better than the sensor accuracy provided by the producer. Our results for the silty clay loam soil in the HOAL catchment are consistent with the findings from Kovács et al. (2019) who evaluated the performance of 28 FP sensors in the laboratory using a different approach for four other soil types: sandy loam, clay loam, loam, and loamy sand. Although differences were observed for the four soil types the

authors report the following similar behaviour of the FP sensors: positive deviation from actual water content in dry condition with highest deviation for very dry soils, which coincides with our findings. The authors further report a negative deviation for water content above 0.40 m$^3$m$^{-3}$. We cannot observe this negative deviation from the gravimetric water content as the high clay content of the silty clay loam soil from the HOAL catchment prohibits a manual preparation of a water content higher than 0.30 m$^3$m$^{-3}$ and full saturation is reached at the water content of 0.48 m$^3$m$^{-3}$.

In order to establish a calibration function for the FP sensor observations in the HOAL catchment, a linear function was fitted





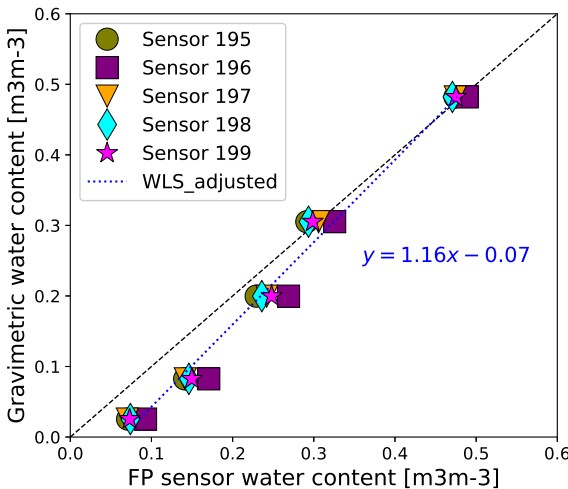

**Figure 3.** Sensor readings are plotted against gravimetric water content under laboratory conditions. The fitted function, obtained by applying the weighted least squares (WLS) method, is displayed as a dotted line (WLS_adjusted).

through the observations obtained in the laboratory experiment based on the weighted least squares method (Fig. 3). Following this method, the weights of the linear regression for the different water content levels were calculated as the inverse variances of the five FP sensors (Table A2). Only the regression weight of the water content level of 0.30 $m^3m^{-3}$ was divided in half and set to 0.3 as we assume that the sensor readings are not fully reliable. At this specific water level the clay particles started to

aggregate, due to the high clay content in the soil, and small gaps of air developed during the process of manually mixing the soil. As a consequence, it is likely that the FP sensors underestimated the actual water content. Thus, the lower weight was given to the observations at this gravimetric water content. The resulting equation of the fitted function

$$SM_{cal} = 1.16 SM_{FP} - 0.07 \, [m^3 m^{-3}], \tag{4}$$

where $SM_{FP}$ represents the FP sensor soil moisture observation and $SM_{cal}$ stands for the calibrated soil water content, is

valid for silty clay loam and was applied to all observations obtained by the FP sensors in the HOAL catchment.

## 4.2 Comparison with scientific probes

### 4.2.1 Temporal agreement

**Air temperature**






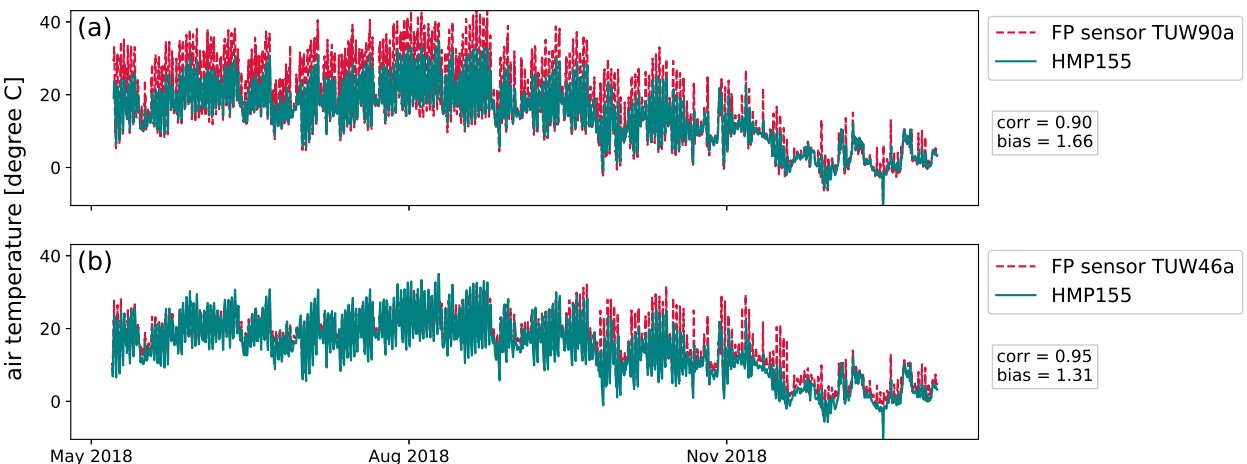

**Figure 4.** (a): Air temperature observed with the professional probe HMP 155 and the FP sensor TUW90a, both located at the weather station in the middle of the HOAL catchment. (b): Air temperature observed with the FP sensor TUW46a located in the sparse forest and the professional probe HMP 155 located in grassland at the weather station.

Air temperature measured by five different FP sensors shows high correlation values with the professional HMP 155 instrument, ranging from 0.88 to 0.98, but a considerable RMSD of 2.59 to 5.33 °C and bias of 1.30 to 2.82 °C. This large deviation in absolute values can be explained by the fact that the air temperature sensor implemented within the FP device is heavily warmed by incoming solar radiation, heating up the plastic cover of the sensor and falsifying the reading of the temperature

sensor which sits right below the plastic cover. This temperature effect is apparent from recorded temperature observations exceeding plausible values measured with professional devices, especially during the summer months, when the sun reaches its maximum strength at this latitude (Fig. 4a).

Figure 4 (bottom) compares air temperature observations of the FP sensor TUW46a, located in the sparse forest, with the professional air temperature sensor, installed in grassland cover at the weather station. The protective effect of the vegetation

canopy during the summer months is clearly visible. In the beginning of May, when the leaves in the sparse forest are not fully grown, the sensor is still partly reached and warmed by direct sun light, thus leading to an overestimation of the air temperature. As soon as the canopy cover of the deciduous trees is fully developed and the FP sensor is shaded, the air temperature of the FP sensors corresponds well with the measurements of the professional probe. When leaf fall starts in October and the FP sensor is not protected from the incoming radiation anymore, the deviation from the professional probe is apparent again.

Figure 5a illustrates the daily temperature ranges of the professional air temperature sensor HMP155 and the FP sensors TUW90a with respect to their daily maximum observation. The warming effect of the FP sensor caused by the incoming solar radiation is clearly reflected in the higher maximum values reached by the low-cost sensor. In Fig. 4, it is visible that sensor TUW90a records lower minimum values than the professional sensor, which can be confirmed by comparing the recorded





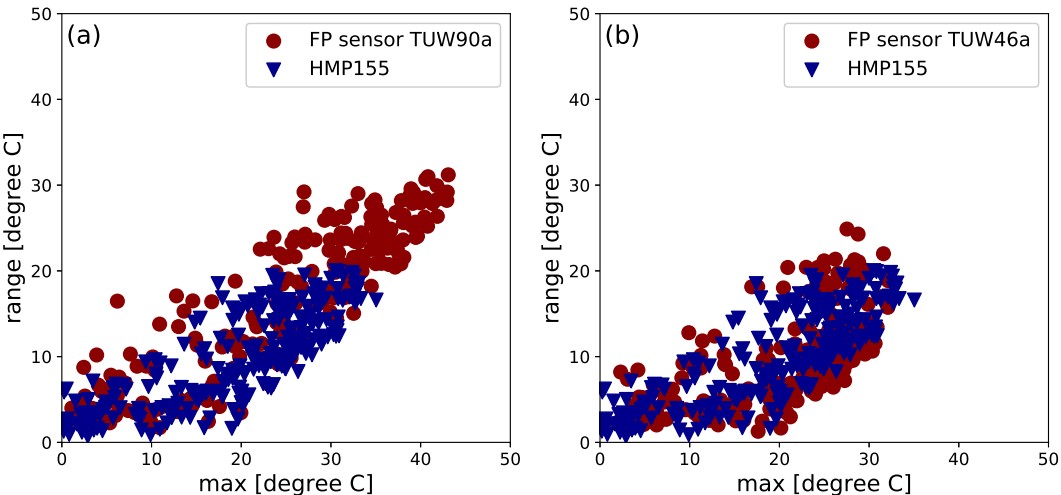

**Figure 5.** Scatter plots of the maximum temperature against the temperature range based on the conventional probe HMP155 and the FP sensor TUW90a (a), and the FP sensor TUW46a (b).

minimum temperatures of boths probes (Fig. A1a). It is unclear if this offset is a general characteristic of the FP temperature

probe. Sensor TUW46a shows a contrasting behaviour (Fig. A1b), recording consistently higher daily minimum values than the HMP155 probe, which could be a result of the different sensor location, TUW46a is located in the sparse forest while HMP155 and TUW90a are installed at the weather station. The variation of daily temperature range (Fig. 5b) is comparable to the daily variation recorded by the conventional probe. Only between daily maximum temperatures of 25 and 30 °C the daily range of the FP sensor varies more than for the HMP155, which is probably a result of the temperature effect discussed

earlier. Overall, a high agreement with the professional air temperature device was reached indicating that the ambient temperature measurements of the FP sensor can be beneficial for environmental applications. Certainly, the warming effect of the FP sensor's plastic cover due to incoming solar radiation has to be considered. The light intensity observed by the FP sensor could serve as a reference to filter out untrustworthy temperature observations. If the absolute temperature values are of interest further investigations are recommended as our initial investigation showed a possible offset from observations obtained with

the professional probe.

**Light level**

Figure 6 shows the light intensity observed by the FP sensor TUW90a, installed in 1.7 m height, in relation to the down-

welling shortwave radiation obtained from the professional sensor CNR4 Net-Radiometer, both located at the weather station of the HOAL catchment. Despite the much smaller range of incoming shortwave radiation observed by the FP sensors than the professional probe, i.e. only the visible domain, a reasonable agreement underpinned by a correlation value of $0.87$ is





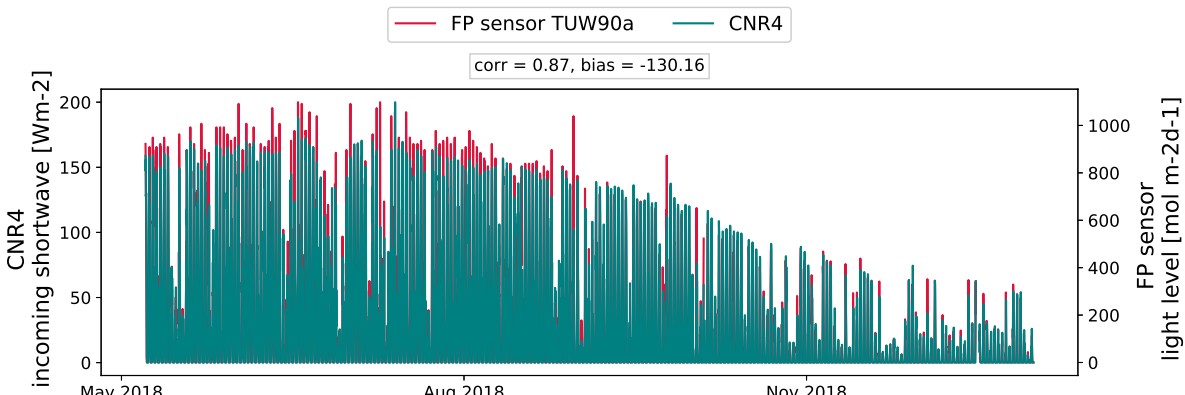

**Figure 6.** Light level variable observed with the FP sensor TUW90a expressed in mol m$^{-2}$d$^{-1}$ plotted against the incoming shortwave radiation in Wm$^{-2}$ from the professional probe CNR4.

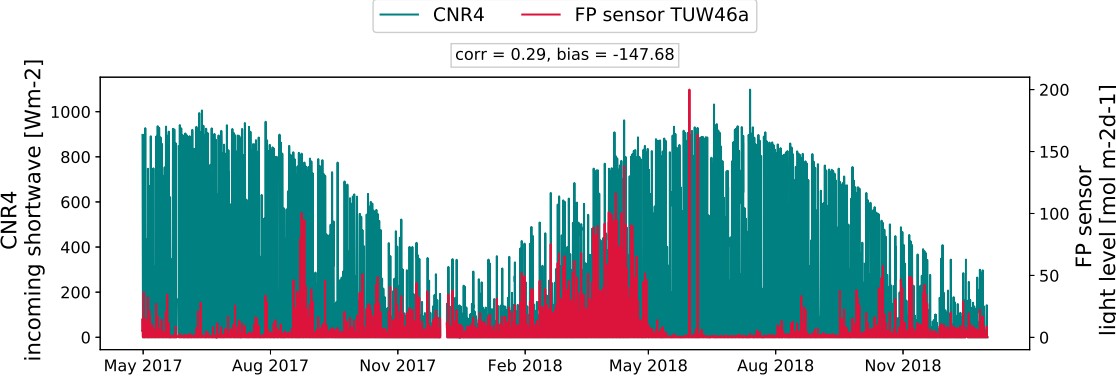

**Figure 7.** Same as Fig. 6, but for FP sensor TUW46a (located in sparse forest).

reached. The high deviation in absolute values is a consequence of the different observation ranges and measurement units of both devices, which cannot be easily transformed into a common unit due to the difference in the observed spectral range.

A variation of the incoming radiation throughout the seasons can be observed for both devices. In Fig. 7 this annual cycle is clearly visible in the observations of the CNR4 probe. For the FP sensor TUW46a, located in sparse forest, the effect of fully developed canopy cover during the summer season shading the sensor is evident, resulting in a low correlation value of $0.29$ with the professional probe, which is located in the open field.

The light intensity observations of the FP probes are certainly useful to track the development of canopy cover, i.e. when the sensor is shaded, and to provide a reliability measure for the temperature measurements. Due to the lack of a professional




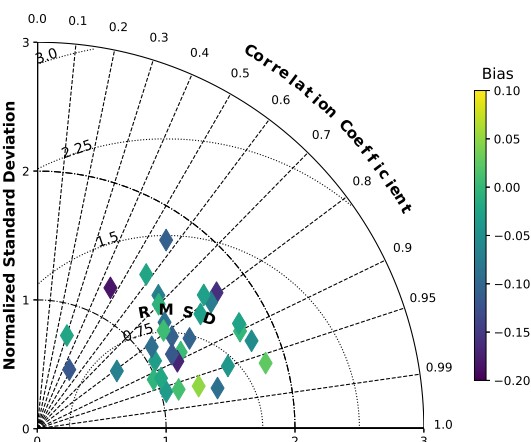

**Figure 8.** Taylor diagram showing agreement between FP sensors and professional probes at 5 cm depth.

sensor with the same observational wavelength range we are not able to provide a more detailed analysis. Thus, for any further applications we recommend to compare the FP sensor observations with an appropriate professional device.

### Soil moisture

For the sensor-to-sensor comparison between the FP sensors and the professional probes at 0.05 m depth an overall correlation coefficient of 0.80 was obtained with individual correlation coefficients ranging from 0.30 to 0.97. The average bias is $-0.05$ m$^3$m$^{-3}$ and the overall RMSD is 0.08 m$^3$m$^{-3}$ (uRMSD 0.05 m$^3$m$^{-3}$). For the comparison with the professional sensors at 0.10 m depth an overall correlation of 0.79, a bias of $-0.06$ m$^3$m$^{-3}$, and a RMSD of 0.10 m$^3$m$^{-3}$ (uRMSD 0.05 m$^3$m$^{-3}$) were observed. The results are visualized in Taylor diagrams (Taylor, 2001) in Fig. 8 and Fig. A7.

The FP sensors and professional probes agree well in both depths. As the results compared with the professional probes at 0.05 m and 0.10 m depth are in a very similar range and are driven by similar characteristics, only the comparison with the near surface layer is discussed in more detail.

Figure 9 serves as an example of the overall good agreement between a FP sensor and a professional probe. For this location, precipitation events are captured simultaneously by both sensor types resulting in synchronous time series shapes and a high correlation value of 0.96. In comparison with the scientific probe the FP sensor is showing higher noise (0.005 m$^3$m$^{-3}$) and intra-daily variability (0.015 m$^3$m$^{-3}$), which is most likely a sensitivity to temperature resulting from the measurement frequency, both effects are well below the sensor accuracy provided by the manufacturer. In general, the measurements of the FP sensors are characterized by pronounced responses to precipitation events, while often much weaker or in some cases no reaction is recorded by the professional probes. This different reaction strength causes the standard deviation of the FP sensors to be considerably higher than the standard deviation of the professional probes, in the case of sensor TUW267a even up to 2.2 as





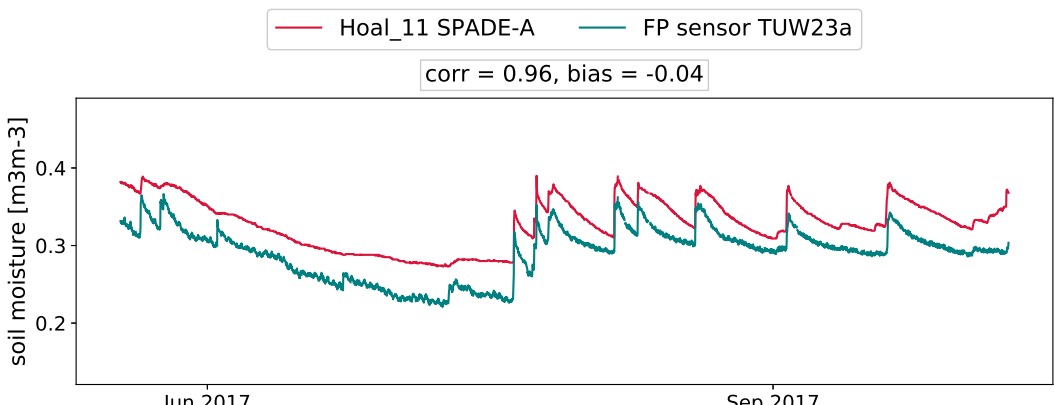

**Figure 9.** Soil moisture time series of FP sensor TUW23a and professional probe Hoal_11 at 0.05 m depth, installed next to each other, in good agreement.

high (Fig. A2). The considerable difference in absolute soil moisture values, visible in the overall results (Fig. 8), can also be explained by the vertical installation of the FP sensors. The low-cost devices record the soil water content from the surface, which is wetting and drying out fast, down to a depth of ten centimeters. The horizontally installed professional probes are less affected by the drying surface and in general record higher water content levels. In addition, differences in the absolute water content can be caused by local variations in soil texture. Although the FP devices were calibrated for the dominant soil texture type present in the HOAL catchment, the conditions in the laboratory are ideal (e.g. homogeneous soil sample) and less perfect conditions with local variations in soil texture are to be expected in the field. Another reason for observed instances of poor correspondence, represented by low correlation values, is suspicious behaviour or sensor failure of one of or both sensors, e.g. extreme response to negative temperatures (e.g., Fig. A4).

The soil moisture readings of the two FP sensors buried in a horizontal position at a depth of 0.05 m agree very well with the observations of the horizontally installed professional probes (e.g. Fig. A3). High correlation values of 0.96 and 0.97 are reached. In addition, the deviation in absolute values is low with the bias varying from 0.01 to 0.04 and RMSD from 0.02 to 0.05 $m^3m^{-3}$ (uRMSD from 0.02 to 0.03 $m^3m^{-3}$). This observation supports the hypothesis that part of the discrepancies observed for the other sensors stem from the differences in installation (i.e., vertical vs. horizontal).

So far we have demonstrated the performance of the FP sensors in comparison with the TDT SPADE sensors. Now, we investigate the results with respect to the scientific probe 5TM. The two sensors, both based on the capacitance technology, show a good temporal agreement with a correlation value of 0.89 (Fig. A5). Clearly visible is the broader value domain of the low-cost soil moisture observations, which results in a RMSD of 0.06 $m^3m^{-3}$ and a normalized standard deviation of 1.77, and could be a consequence not only of the different sensor position but also of different measurement frequencies used by the two sensor types. The statistical metrics obtained by the second FP sensor installed next to the 5TM probe show almost identical





values, confirming the low inter-sensor variability already observed during calibration. No clear dependencies of the FP sensor performance compared with the professional devices to land cover or topography were found (not shown).

The inter-comparison of five pairwise installed FP sensors with an average observation length of eleven months resulted in an overall correlation of 0.87, a bias of 0.01 $m^3m^{-3}$, and a RMSD of 0.08 $m^3m^{-3}$ (uRMSD 0.04 $m^3m^{-3}$) (Fig. A8). Individual correlation values range from 0.55 to 0.98, the individual RMSD values from 0.04 to 0.13 $m^3m^{-3}$ (uRMSD 0.02 to 0.07 $m^3m^{-3}$) and the bias from −0.07 to 0.13 $m^3m^{-3}$.

The noticeable high bias of 0.13 $m^3m^{-3}$ (Fig. A8), which occurs for the sensor pair TUW37a and TUW56a located in the forest, can be explained by their different sensor position. FP sensor TUW56a is installed horizontally in 0.05 m depth and is showing constantly higher water content levels than the vertically installed sensor TUW37a, which might be due to the faster drying near the soil surface or different texture conditions. Nonetheless, their good temporal agreement is underpinned by a correlation value of 0.95 (Fig. A6). In addition to the difference in the overall soil moisture level and the amplitude

during the wetting process, another effect caused by the different sensor position is visible: The sensor installed in a vertical position is showing steeper drying curves, reflecting the faster drying process at the soil surface. This characteristic is even more prominent for the sensor pair installed in sparse forest and is responsible for a high RMSD of 0.10 $m^3m^{-3}$ (uRMSD 0.06 $m^3m^{-3}$). The different strength of the described characteristics of both horizontally installed FP sensors compared with their vertically installed counterparts is most likely driven by differences in soil texture at the two locations in the forest and

sparse forest, respectively.

The lowest correlation result of 0.55, accompanied by a noticeable RMSD of 0.09 $m^3m^{-3}$ (uRMSD 0.07 $m^3m^{-3}$), is obtained for the sensor pair TUW30a and TUW31a, which is located in cropland, and seems to be driven by suspicious observations of sensor TUW30a during the winter season. The same FP sensor is also responsible for the lowest correlation value achieved (0.30) in the comparison with the professional probe at 0.05 m depth (Fig. 8, left).

Overall, the high temporal agreement of the FP sensor soil moisture observations with the readings from professional devices is evident, thus allowing the use of the low-cost sensor scientific applications focusing on temporal dynamics. The clear deviation in absolute terms from conventional sensors is mostly a consequence of the different sensor position, as the improved statistics of the buried FP sensors showed.

### 4.2.2 Temporal stability

Figure 10 shows the calculated mean relative differences and the correlation between individual sensors and the catchment mean for the FP sensors (top) and the corresponding professional sensors at 0.05 m depth (bottom). The FP sensors and corresponding professional probes are sorted by the mean relative difference values of the FP sensors. The error bars refer to the standard deviation of the mean relative difference between the catchment average and the individual sensor observations.

While FP sensor TUW19a was identified as the most representative for the catchment average derived from the low-cost

sensors, for the professional probes Hoal_13 was found to be the most representative. Although both sensors are located in the same land cover class (sparse forest) they are not located next to each other. A similar outcome was obtained for the standard deviation of the mean relative difference. The most representative low-cost and professional sensors, TUW34a and Hoal_30,





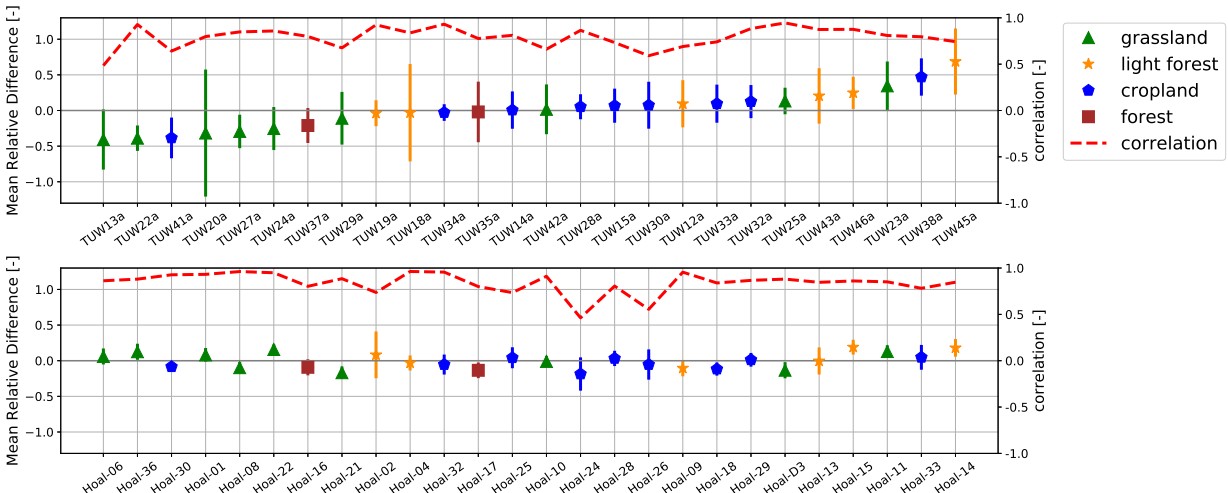

**Figure 10.** Mean relative differences (MRD) for FP sensors (top) and corresponding professional probes at 0.05 m depth (bottom), sorted by the MRD of the FP sensors. In addition, the correlation between the catchment average and individual sensors is shown.

respectively, are both located in cropland, but in different parts of the catchment. In general, a distinct higher variation of and deviation from the mean relative difference of the FP sensors compared with the professional probes is noticeable and is

expected to ba a consequence of the high variance of the water content, driven by the vertical sensor position already discussed in Sect. 4.2.1.

Although known wet locations (area around sensors TUW45a and TUW46a) of the catchment could correctly be identified by the FP sensors as wetter than the catchment average, the known dry area of the catchment, i.e. the forest, is not clearly represented by the FP sensors (TUW35a and TUW37a). Only sensor TUW37a was clearly identified as drier than the network

average. In addition, sensor locations identified as wetter than the catchment average based on the professional sensors were identified as drier than the average when considering the low-cost sensors (e.g. TUW24a and Hoal_22) or vice verca (e.g. TUW13a and Hoal_06). The difference in absolute water content values can also result from local differences in soil texture. Although the FP sensors were installed in close vicinity to the professional probes the exact distance is often unknown (due to different installation dates) but is estimated to be no more than one meter. Consequently, the deviation in absolute values and

mean relative differences cannot solely be reduced to the different sensor types. When looking at the relative agreement with the low-cost sensor based catchment average a considerable high overall correlation of $0.80$ was reached with individual values ranging from $0.48$ to $0.94$ (Fig. 10, Fig. A9). The two sensors, TUW19a and TUW34a, previously identified as most representative are in good agreement with the catchment average when considering the correlation. Noticeably, FP sensor TUW30a is according to the derived mean relative difference a close approximation of the catchment average, but the low correlation value

reveals the existing problems of this sensor, which we already discussed in the previous section. The correlation results based





on the data of the professional sensors are within a comparable range, from $0.46$ to $0.96$ and an overall correlation value of $0.84$. Again, for both sensor types sensors at different locations in the catchment were identified as representing the catchment average the best: TUW25a of the low-cost sensors, located in grassland, and the professional sensor Hoal_04 in sparse forest, respectively. No clear dependency of relative agreement with the catchment average to the land cover is visible.

In summary, the temporal stability results of the FP sensors clearly differ from those of the professional probes and are thus unsatisfactory. Although it has to be recognized that the mean relative difference is strongly affected by absolute soil moisture values, which themselves can strongly be influenced by local variation in soil texture.

### 4.2.3   Soil Moisture Memory

Figure 11 shows the autocorrelation functions and e-folding times $\lambda$ for the catchment average based on the FP sensors and
the professional sensors at 0.05 m depth, respectively. The resulting e-folding time for the FP sensors is with 83 hours reached seven hours earlier than the e-folding time of the professional probes. This is most likely a result of the vertical installation position of the FP sensors, causing pronounced reactions to precipitation events as shown and described in Sect. 4.2.1. While the horizontally installed professional probes show a weaker wetting and drying signal after rain events, causing the autocorrelation curve to be slightly flatter and reaching the e-folding time a few hours delayed. On average an e-folding time of about 3.5 days
was estimated by both sensor types which is in accordance with e-folding times of in situ observations in existing literature (Ghannam et al., 2016).

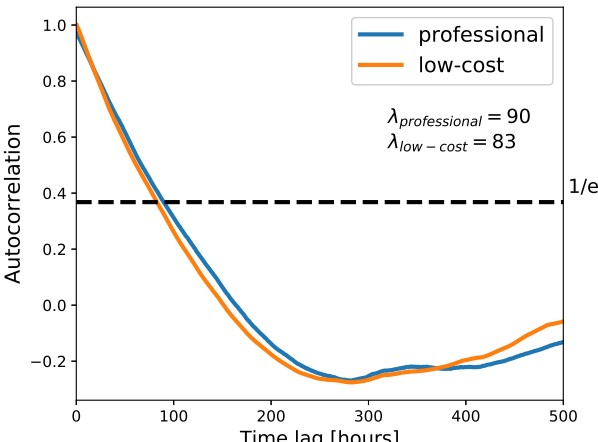

**Figure 11.** Autocorrelation function and e-folding times $\lambda$ for the catchment average based on the FP sensors and the professional probes at 0.05 m depth.

When investigating the e-folding times with respect to the different land cover classes (see Table 2), we find similar results. For three of the four land cover classes the obtained e-folding times are reached earlier for the low-cost sensors than for the





professional probes. The only exception is the land cover class sparse forest, for which a persistence of soil moisture anomalies
four hour longer was identified for the FP sensor observations. For both sensor types the sparse forest was the land cover
class with the shortest soil moisture memory. This can be explained by the known wet locations being situated in sparse forest
(area around sensors TUW45a and TUW46a). Rahman et al. (2015) and Orth and Seneviratne (2013) found a faster decay of
soil moisture memory in wet regions than in dry areas, explained by usually smaller anomalies in the already wet soil. The
same effect can be valid on a catchment scale. This is in agreement with the longer persistence of anomalies in the forest, the
known dry area of the catchment. As already visible in the temporal stability results the forest was identified as dry based on
the professional sensors, but less clear for the FP sensors. Consequently, a stronger memory in the forest is reflected by the
professional probes.

**Table 2.** E-folding times for FP and professional sensors in dependence of the different land cover types.

| land cover type | e-folding time FP sensors [hours] | e-folding time professional sensors [hours] |
|:---:|:---:|:---:|
| cropland | 90 | 91 |
| grassland | 86 | 93 |
| sparse forest | 70 | 66 |
| forest | 88 | 100 |

### 4.2.4   Automated Quality Control

Overall, 24.6% of all soil moisture observations obtained by the FP sensors were flagged according to the ISMN quality proce-
dures, whereas 19.7% of the measurements from professional sensors at 0.05 m depth were flagged. The flagged observations
can be subdivided in different sub-categories (Table 3).

The majority of observations identified as suspicious consists of geophysical consistency identifiers (D01–D05), for both the
FP and professional sensor observations. Especially the temperature related quality indicators (D01–D03) are merely a conse-
quence of the climate prevalent at the HOAL catchment, which is characterized by pronounced winter seasons with tempera-
tures below 0 °C, and not a sign for suspicious sensor behaviour. Naturally, air reaches freezing temperatures more frequently
than soil. Thus, the percentage of air temperature-dependent flagged observations is much higher for the FP probe that measure
air temperature than for the scientific devices that measure soil temperature. A considerably larger percentage of soil moisture
observations during periods with negative GLDAS soil temperature (D03) is noticeable for the professional sensor than for the
low-cost devices. This difference results from a 20 day data gap for all SPADE sensors in the HOAL catchment in spring 2018,
caused by a damaged fiber optic cable. Both sensor types also recorded a considerable amount of values that exceeded the
saturation point (identifier C03), derived from the Harmonized World Soil Database (FAO/IIASA/ISRIC/ISSCAS/JRC, 2012).
As we are already aware of the high sensitivity of the FP sensors to rain events, driven by their vertical sensor position, we are





**Table 3.** ISMN quality identifiers, their description, and the percentage of flagged values in each class (as percentage of total number of flagged observations), both for the low-cost and professional sensors.

| ISMN quality identifier | FP sensor observations [%] | Professional sensor observations [%] | ISMN quality identifier description |
|---|---|---|---|
| C01 | 0 | 0 | soil moisture $< 0.0$ m$^3$m$^{-3}$ |
| C02 | 0 | 2.9 | soil moisture $> 0.6$ m$^3$m$^{-3}$ |
| C03 | 17.3 | 14.2 | soil moisture $>$ saturation point |
| D01 | 0 | 0.7 | in situ soil temperature $< 0°$C |
| D02 | 16.4 | 0 | in situ air temperature $< 0°$C |
| D03 | 40.0 | 64.3 | GLDAS soil temperature $< 0°$C |
| D04 | 4.0 | 0 | soil moisture shows peaks without in situ rain event in preceding 24 hours |
| D05 | 5.7 | 10.4 | soil moisture shows peaks without GLDAS rain event in preceding 24 hours |
| D06 | 0.0 | 0.0 | a spike is detected in the soil moisture spectrum |
| D07 | 0.0 | 0.0 | a negative break is detected in the soil moisture spectrum |
| D08 | 0.1 | 0.0 | a positive break is detected in the soil moisture spectrum |
| D09 | 11.9 | 0.4 | low constant values occur in the soil moisture spectrum |
| D10 | 4.6 | 7.1 | a saturated plateau occurs in the soil moisture spectrum |

not surprised about the higher percentage of soil moisture observations above the saturation point. Pronounced differences in the flagging statistics can be observed for the following ISMN quality indicators: D09, D10, and C02. With exception of D09, a higher percentage of flags was identified for the professional sensor observations. The D09 flag identifies a period of measurements with very low variability following a distinct negative break. On the one hand, this flag is detected in FP observations during winter season, when the soil moisture drops due to low temperatures and remains at a low level. Consequently, the D02 identifier is also attached to these observations. On the other hand, it was observed that soil moisture values drop to zero and remain there for some time without any visible physical damage of the FP sensor. The reason is unknown, but we speculate that this is a consequence of corroding contacts.

Measurements detected outside the geophysical plausible range of 0.60 m$^3$m$^{-3}$ capture a problem prevalent for the professional sensors. Soil moisture suddenly rises to unrealistic high values and remain at this level for some time. We speculate that this is caused by an energy supply problem or a connection problem to the data logger. The same suspicious measurements are also responsible for a higher amount of D10 identifiers than observed in the FP observations. Although the underlying algorithm for identifying suspicious positive breaks (D08) caught only a small percentage of FP sensor observations, it is worth mentioning that mainly precipitation-induced raises in soil moisture triggered this quality identifier.

Overall, the ISMN quality flagging statistics underpin that the low-cost sensors provide meaningful observations comparable to those from professional probes. The main sensor characteristic, the high sensitivity to precipitation due to the vertical





installation position, was captured by the quality identifiers. For any scientific usage the FP sensor flaw of unexplained soil
moisture values of zero, revealed by the D09 flag, has to be considered by masking the zero values. Although the amount of
these zero values can take up a large portion of a sensor's time series, fortunately this problem occurred only for a very limited
number of FP sensors (three out of 52). Thus, this problem can be overcome, either by installing multiple low-cost sensors at
the same location or during data analysis by removing those observations retrospectively.

### 4.2.5    Example: Scientific application

To demonstrate the suitability of the FP sensors for scientific applications, we evaluated the Advanced SCATterometer (ASCAT)
remotely-sensed soil moisture product (H113, H114) at the HOAL catchment by using the low-cost and the professional sen-
sors. The ASCAT instruments on-board the Meteorological Operational (Metop) satellite series provide observations with a
spatial resolution of 25 km and a revisit time of every 1–2 days. The retrieved soil moisture product (Wagner et al., 1999;
Naeimi et al., 2009; Wagner et al., 2013) represents soil moisture in the upmost layer of the soil and is given in a relative
unit as degree of saturation. To overcome the mismatch of units and spatial resolution between the satellite product and the in
situ sensors, the satellite-based observations were re-scaled to the mean and standard deviation of the ground data. In order to
ensure longest possible observation periods of the ground observations only permanently installed sensors were used to derive
the catchment average. Daily means of water content were calculated and disregarded if air temperature values were below
3 °C. For the FP sensors this filtering was done based on their air temperature records, for the professional probes air tempera-
490    ture observations from the weather station in the middle of the HOAL catchment were applied. In addition to the absolute soil
moisture values, we compared the anomalies of the satellite and the ground data, which we derived by using a 35 day moving
average.

**Table 4.** Evaluation results of the ASCAT soil moisture product at the HOAL catchment with respect to the FP sensor and professional sensor
data. Results are given for the soil moisture observations and the soil moisture anomalies.

| Metric | FP sensors | | Professional sensors | |
|---|---|---|---|---|
| | soil moisture | anomalies | soil moisture | anomalies |
| correlation [-] | 0.75 | 0.42 | 0.76 | 0.26 |
| uRMSD [$m^3 m^{-3}$] | 0.03 | 0.02 | 0.02 | 0.02 |

The evaluation results are shown in Table 4. The correlation between the satellite product and the ground observations based
on the absolute soil moisture values is high and almost identical with $0.75$ and $0.76$ for the low-cost and professional sensors,
495    respectively. The same is true for the uRMSD with $0.03$ $m^3 m^{-3}$ for the FP sensors and $0.02$ $m^3 m^{-3}$ for the scientific sensors.
When considering the soil moisture anomalies, the uRMSD is identical for both sensor types. A much weaker agreement
between the anomalies of the satellite product and the sensors was obtained with correlation values of $0.42$ and $0.26$, where the
higher correlation value was achieved with the low-cost values.





These results herewith demonstrate the suitability of the FP sensors for satellite validation. It is expected that many other
scientific applications can benefit from the use of low-cost soil moisture sensors, i.e. the FP sensor, and a variety of scientific
experiments become possible only due to the use of low-cost devices.

## 5 Conclusions

In this study we investigated the performance and scientific suitability of a commercial low-cost soil moisture sensor in the
laboratory and in the field, which lead us to the following conclusions:

- Laboratory calibration showed that the FP sensor overestimates soil water content in dry conditions of the silty clay loam
  soil of the Austrian catchment. Similar to professional sensors a site specific calibration is highly recommended if the
  absolute amount of soil moisture is of interest. For applications where relative measurements have priority, i.e. satellite
  validation, FP sensors are very well suitable without laboratory calibration.

- In addition, we investigated the inter-sensor variability of the FP sensor in the laboratory and found very small differences
  between the sensors, which confirms their reliability.

- The comparison with professional soil moisture sensors in the field shows a high agreement and confirms their ability
  to capture the wetting and drying process in the soil. Discrepancies were mostly driven by the different sensor position
  (vertical/horizontal) and local variations in soil texture.

- The strongest weakness of this sensor, of recording zero soil moisture values without any physical damage, was ob-
  served only for a very limited number of devices and can be overcome by a clever experiment set-up. The compactness,
  simplicity, and affordable price of the sensor allows for installation of multiple devices at the same location and easy
  re-installation or replacement. In general, these characteristics hold the potential for many scientific applications with a
  large number and high density of sensors otherwise not feasible with professional sensors, e.g. (Zappa et al., in prepara-
  tion, i).

- Field testing has proven that the FP sensor can sustain outdoor conditions. In fact, the longest surviving FP sensor within
  the framework of GROW is still recording reasonable data after more than 2.5 years in the field. This shows that the FP
  sensor can be deployed for short- and medium-term applications, while long-term usability still has to be investigated.

Overall, we conclude that the FP sensor is suitable for scientific environmental applications and we expect that many scien-
tific studies will benefit from the use and abilities of the FP sensor data collected within the GROW project. The deployment of
such a low-cost sensor within a citizens' observatory can provide an effective mechanism to support current Earth observation
capabilities.





*Data availability.* The in situ soil moisture observations are available through University of Dundee and will become available in the ISMN (International Soil Moisture Network).





## Appendix A

**Table A1.** Available FP sensors used for temporal stability analysis.

| FP sensor | data available from | data available to | sensor position | land cover | elevation a.s.l. [m] | professional sensor |
|---|---|---|---|---|---|---|
| TUW12a | 2017-04-30 | 2018-10-25 | vertical | sparse forest | 293.97 | Hoal_09 |
| TUW13a | 2017-04-30 | 2018-08-27 | vertical | grassland | 282.56 | Hoal_06 |
| TUW14a | 2017-04-30 | 2018-08-28 | vertical | cropland | 277.15 | Hoal_25 |
| TUW15a | 2017-04-30 | 2018-08-28 | vertical | cropland | 280.36 | Hoal_28 |
| TUW18a | 2017-04-30 | 2018-10-25 | vertical | sparse forest | 305.47 | Hoal_04 |
| TUW19a | 2017-04-30 | 2018-10-25 | vertical | sparse forest | 309.69 | Hoal_02 |
| TUW20a | 2017-04-30 | 2018-10-25 | vertical | grassland | 321.06 | Hoal_01 |
| TUW22a | 2017-04-30 | 2018-11-16 | vertical | grassland | 289.27 | Hoal_36 |
| TUW23a | 2017-04-30 | 2018-10-25 | vertical | grassland | 289.77 | Hoal_11 |
| TUW24a | 2017-04-30 | 2018-10-25 | vertical | grassland | 281.79 | Hoal_22 |
| TUW25a | 2017-04-30 | 2018-11-23 | vertical | grassland | 274.67 | Hoal_D3 |
| TUW27a | 2017-04-30 | 2018-11-23 | vertical | grassland | 274.93 | Hoal_08 |
| TUW28a | 2017-04-30 | 2018-09-04 | vertical | cropland | 284.71 | Hoal_24 |
| TUW29a | 2017-04-30 | 2018-09-04 | vertical | grassland | 289.77 | Hoal_21 |
| TUW30a | 2017-04-30 | 2018-07-26 | vertical | cropland | 272.86 | Hoal_26 |
| TUW32a | 2017-04-30 | 2018-07-26 | vertical | cropland | 274.01 | Hoal_29 |
| TUW33a | 2017-04-30 | 2018-07-26 | vertical | cropland | 292.21 | Hoal_18 |
| TUW34a | 2017-04-30 | 2018-02-14 | vertical | cropland | 292.21 | Hoal_32 |
| TUW35a | 2017-04-30 | 2018-08-01 | vertical | forest | 280.13 | Hoal_17 |
| TUW37a | 2017-04-30 | 2018-11-23 | vertical | forest | 279.04 | Hoal_16 |
| TUW38a | 2017-04-30 | 2018-07-27 | vertical | cropland | 279.22 | Hoal_33 |
| TUW41a | 2017-04-30 | 2018-09-04 | vertical | cropland | 292.21 | Hoal_30 |
| TUW42a | 2017-04-30 | 2018-11-16 | vertical | grassland | 292.21 | Hoal_10 |
| TUW43a | 2017-04-30 | 2018-08-27 | vertical | sparse forest | 278.20 | Hoal_13 |
| TUW45a | 2017-04-30 | 2018-11-23 | vertical | sparse forest | 271.08 | Hoal_14 |
| TUW46a | 2017-04-30 | 2018-11-23 | vertical | sparse forest | 271.70 | Hoal_15 |





**Table A2.** Gravimetric soil water content (first column), soil water content measured with FP sensors (columns two to six), standard deviation of FP sensors (column seven) and weights for the weighted least squares (WLS) fit (column eight).

| Water content [m³m⁻³] | Sensor 195 [m³m⁻³] | Sensor 196 [m³m⁻³] | Sensor 197 [m³m⁻³] | Sensor 198 [m³m⁻³] | Sensor 199 [m³m⁻³] | Standard deviation [m³m⁻³] | Weights for weighted least squares fit |
|---|---|---|---|---|---|---|---|
| 0.0255 | 0.0706 | 0.0927 | 0.0703 | 0.0740 | 0.0731 | 0.008 | 1.417 |
| 0.0825 | 0.1409 | 0.1706 | 0.1417 | 0.1461 | 0.1497 | 0.011 | 0.846 |
| 0.1997 | 0.2290 | 0.2689 | 0.2420 | 0.2360 | 0.2480 | 0.014 | 0.540 |
| 0.3053 | 0.2914 | 0.3257 | 0.3059 | 0.2935 | 0.2987 | 0.012 | 0.300 |
| 0.4821 | 0.4739 | 0.4889 | 0.4743 | 0.4708 | 0.4751 | 0.006 | 2.503 |

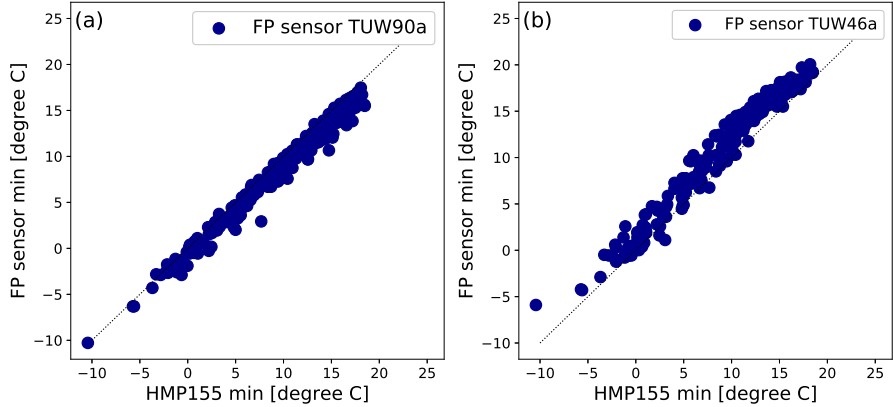

**Figure A1.** Scatter plots of the minimum temperatures observed with the conventional probe HMP155 and the FP sensors TUW90a (a) and TUW46a (b).





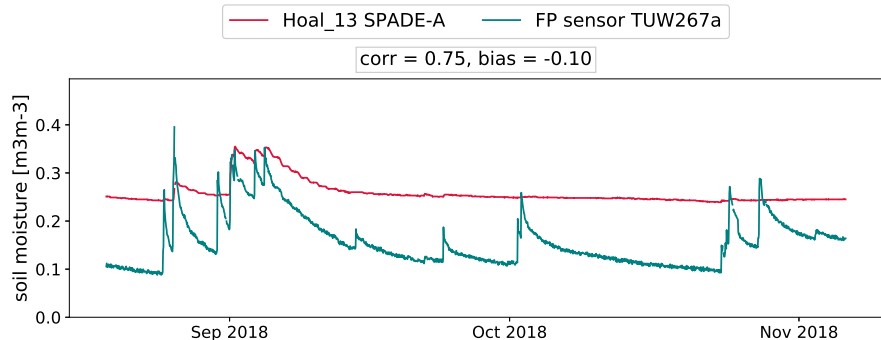

**Figure A2.** Soil moisture time series of FP sensor TUW267a and professional probe Hoal_13 at 0.05 m depth, installed next to each other. While the low-cost sensor shows distinct peaks due to rain events, the professional probe reacts considerably weaker.

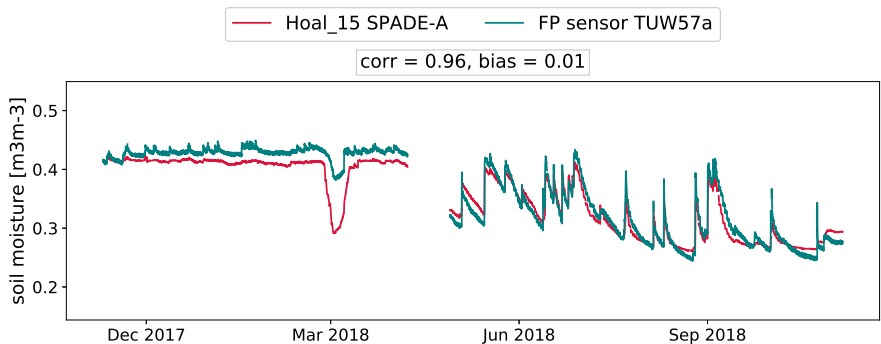

**Figure A3.** Soil moisture time series of horizontally installed FP sensor TUW57a and professional probe Hoal_15 at 0.05 m depth, installed next to each other. The good agreement is underpinned by a high correlation value and low bias.

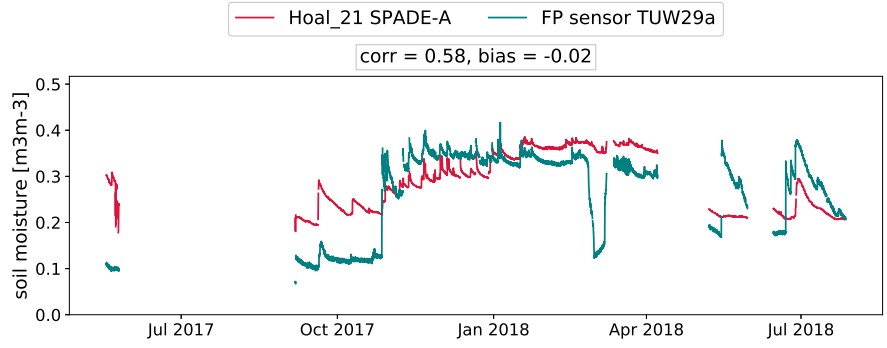

**Figure A4.** Soil moisture time series of FP sensor TUW29a and professional probe Hoal_21 at 0.05 m depth, installed next to each other.





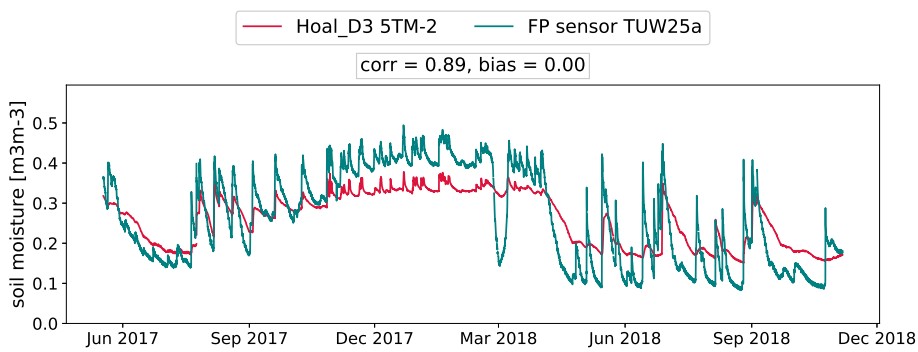

**Figure A5.** Soil moisture time series of FP sensor TUW25a and professional probe Hoal_D3 at 0.05 m depth, installed next to each other.

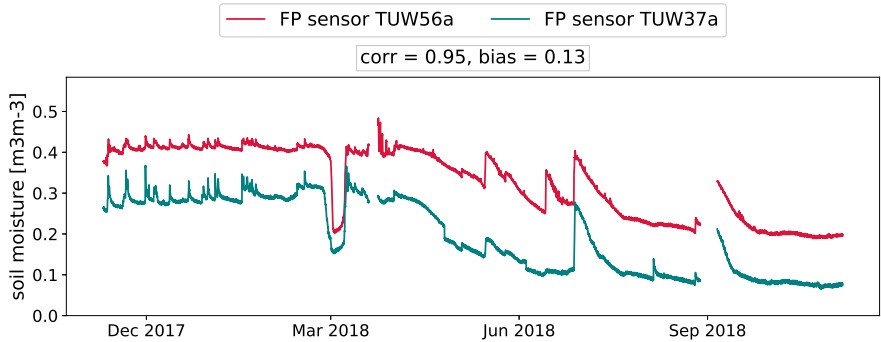

**Figure A6.** Soil moisture time series of FP sensors number TUW37a and TUW56a, where TUW37a is installed vertically and TUW56a horizontally.





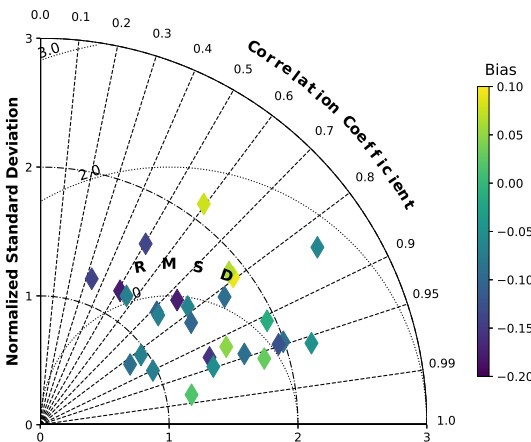

**Figure A7.** Taylor diagrams showing agreement between FP sensors and professional probes at 10 cm depth.

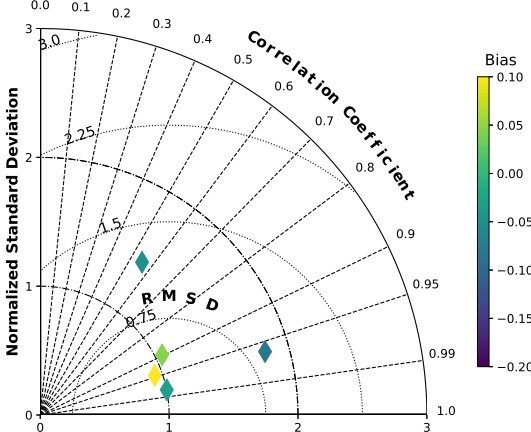

**Figure A8.** Taylor diagram showing agreement between pairwise located FP sensors.





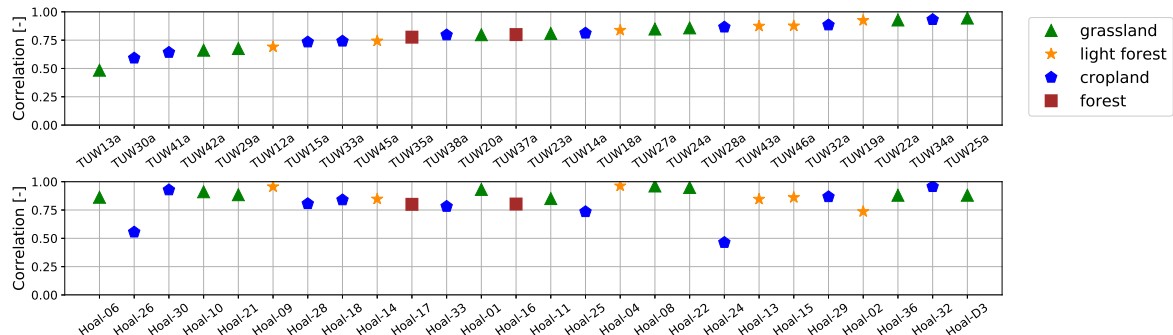

**Figure A9.** Correlation between the catchment average and individual sensors for FP sensors (top) and corresponding professional probes at 0.05 m depth (bottom), sorted by the correlation value of the FP sensors.



*Author contributions.* A.X. conceived and designed the experiments; A.X. analyzed the data and wrote the paper; A.X. and L.Z. collected the FP sensor measurements with support of G.R.; A.X. conducted the laboratory calibration with help of I.P. and L.Z.; I.P. and M.V. provided the soil moisture data of the professional sensors; G.R. provided data of all other used professional sensors; and D.H. and W.D. contributed with their expertise. All authors participated in the revision of the manuscript.

*Competing interests.* The authors declare that they have no conflict of interest.

*Acknowledgements.* This project has received funding from the European Union's Horizon 2020 research and innovation programme under grant agreement No 690199, the GROW Observatory (https://growobservatory.org/), and from the International Soil Moisture Network (https://ismn.geo.tuwien.ac.at/) under IDEAS+ of the European Space Agency (contract no. TVUK/AG/18/02082). Wouter Dorigo acknowledges the TU Wien Wissenschaftspreis 2015. In addition, the authors acknowledge the TU Wien University Library for financial support through its Open Access Funding Program. We want to thank the entire team of the Institute for Land and Water Management Research (Federal Agency of Water Management) in Petzenkirchen, Austria, for their kind support. Especially, Monika Kumpan for sharing her expertise in the matter of sensor calibration and her support during the laboratory evaluation, Matthias Oismüller for his support with the data collection in the field and Peter Strauss for granting access to all facilities of the Institute in Petzenkirchen and his general support.



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

ground measurements, Remote Sensing, in review.