# Peer review of "Evaluating the suitability of the consumer low-cost Parrot Flower Power soil moisture sensor for scientific environmental applications"

_Geoscientific Instrumentation, Methods and Data Systems, 2019_

## Referee Comment (RC1) · Anonymous Referee #1 · 9 Dec 2019

I found the manuscript really interesting and pleasant to read. My major concern regards the calibration of FP sensor (par. 3.1). According to what I understood, the Authors took only one soil sample, divided it in 5 parts, with different degrees of sauration, and procedeed to a comparison between the measures of soil moisture provided by 5 FP sensors and 1 professional sensor. The results of the calibration are depicted in Figure 3. Based on my experience on soil moisture data and measurements provided by professional sensors (TDR in particular) I find this approach particularly risky, in particular because the conditions of soil aggregation (macro pores and other) can strongly change from point to point, also in small scales. I suggest the Authors to elaborate the concept and stress the calibration paragraph, highlighting the need of a

more complete (in terms of soil samples) comparison between the sensors. Somehow, I have the feeling that the differences reported in Figure 9 could be attributed to what aforementioned. Morevoer, I believe the Authors should be more cautious in describing the obtained results in terms of soil moisture also because some errors could have been determined due to the difference in the spatial position between the FP and the professional probes. Indeed the Authors state that "The exact horizontal distance between the FP sensors and the professional probes is unknown...", and that it could be approximately 1 meter. Based on my experience, natural soils (not in laboratories) can present strong differences also in small distances, in particolare concerning the hydraulic conductivity.

---

## Referee Comment (RC2) · Anonymous Referee #2 · 19 Dec 2019

General comment The manuscript is well written and scientifically solid. The research is of interest and highlights potential applications for both in situ soil monitoring and satellite validation for large scale analysis. To my view the manuscript can be accepted after minor revisions, here listed.

Specific comments: 1) Page 2, line 37. Replace "Ochsner et al. (2013)" with "(Ochsner et al., 2013). 2) Page 7, Line 143. Maybe "where not considered" is "where considered" instead 3) Page 8, lines 188-189. The sentence is redundant as respect to line 138, page 6. 4) Page 13, line 311. "Despite the much smaller range of incoming shortwave radiation observed by the FP sensors..." It seems that Figure 6 represents only the

incoming shortwave radiation from CNR4, while light level is represented by the FP sensor. 5) Page 14, Figure 7 and lines 313-314. The authors should avoid to calculate the bias or deviations of two different variables characterized by different measurement units. Remove the bias value from the Figure 7. Just a comment on the fact that the deviation cannot be calculated because of the different measurement units is fine. 6) Pag 16, Figure 9. A bar plot of the rainfall in the same graph of the soil moisture measurements can be useful to see if the higher noise of FP sensor is related always to rainfall impulses.vThis can be also better justify the sentence in lines 339-341. 7) Page 18, Lines 415-417. The author should test the significance of the correlation coefficients using the critical values related to the sample size or the p-value test. Same for table 4, on the validation of the ASCAT product.

---

## Author Comment (AC1) · 29 Jan 2020

**Reviewer 2**

**General comments**

**The manuscript is well written and scientifically solid. The research is of interest and highlights potential applications for both in situ soil monitoring and satellite validation for large scale analysis. To my view the manuscript can be accepted after minor revisions, here listed.**

*Response:* We thank the reviewer for his/her positive feedback and the recommenda-

tion to publish the manuscript after minor revision.

We appreciate the constructive comments. Below we address the reviewer's comments point by point.

**Specific comments**

**1) Page 2, line 37. Replace "Ochsner et al. (2013)" with "(Ochsner et al., 2013).**
*Response:* We kindly thank the reviewer for pointing out this oversight. We modified the reference accordingly.

**2) Page 7, Line 143. Maybe "where not considered" is "where considered "instead**
*Response:* Thanks for thoroughly reading our manuscript and for identifying this slip of the pen. We modified the sentence as suggested.

**3) Page 8, lines 188-189. The sentence is redundant as respect to line 138,page 6.**
*Response:* While line 138 (page 6) focuses on the sensor set-up (and highlights the unconventional sensor position and its intended purpose), lines 188-189 (page 8) specify sensors that were used to investigate the temporal agreement between the low-cost and the professional sensors. Although we agree that the information provided is redundant, we decided to leave it as is for the benefit of the reader (to make the sensor easily recognizable).

**4) Page 13, line 311. "Despite the much smaller range of incoming shortwave radiation observed by the FP sensors..." It seems that Figure 6 represents only the incoming shortwave radiation from CNR4, while light level is represented by the FP sensor.**
*Response:* It is true that Figure 6 represents the incoming shortwave radiation from

the CNR4 and the light level observed by the FP sensor. While the CNR4 observes the incoming shortwave radiation within a spectral range from 300 to 2800 nm (see lines 106-107 on page 5), the FP sensor observes only a small part of it. Indeed, the FP sensor only observes the visible light within a spectral range from 400 to 700 nm (lines 75-76, page 3).

**5) Page 14, Figure 7 and lines 313-314. The authors should avoid to calculate the bias or deviations of two different variables characterized by different measurement units. Remove the bias value from the Figure 7. Just a comment on the fact that the deviation cannot be calculated because of the different measurement units is fine.**

*Response:* Thank you for this very valid remark. We have removed the bias from the figures 6 and 7 (see below). In lines 313-314 (page 14) we changed the sentence from "The high deviation in absolute values is a consequence of the different observation ranges and measurement units of both devices, which cannot be easily transformed into a common unit due to the difference in the observed spectral range." into "Computing the bias between the observations made by the devices is unfeasible because of the different wavelength ranges observed and the different measurement units.".

**6) Pag 16, Figure 9. A bar plot of the rainfall in the same graph of the soil moisture measurements can be useful to see if the higher noise of FP sensor is related always to rainfall impulses. This can be also better justify the sentence in lines 339-341.**

*Response:* Thank you for this suggestion. We included the rainfall observations in Figure 9 and in addition in Figure A2. While in Figure 9 the professional and the low-cost soil moisture sensor show a very similar reaction to rainfall events (p. 15, lines 335-337), Figure A2 shows an example where sensitivity to rainfall strongly deviates (p. 15, lines 339-343). This difference is most likely driven by the vertical position of the FP sensors, while the magnitude in general depends on local conditions (e.g., soil
texture, vegetation).
We could not identify a clear connection between the strength of noise and rainfall events.

**7)Page 18, Lines 415-417. The author should test the significance of the correlation coefficients using the critical values related to the sample size or the p-value test. Same for table 4, on the validation of the ASCAT product.**
*Response:* We included the significance information on basis of the p-value as suggested in Table 4 and in the lines 328 (page 15), 370 (page 17), 414 (page 18), and 419 (page 19).

[Figure]

**Fig. 1.** Figure 6

[Figure]

**Fig. 2.** Figure 7

[Figure]

**Fig. 3.** Figure 9

[Figure]

**Fig. 4.** Figure A2

---

## Author Comment (AC2) · 29 Jan 2020

**Reviewer 1**

**General comments**

**I found the manuscript really interesting and pleasant to read. My major concern regards the calibration of FP sensor (par. 3.1).**
**According to what I understood, the Authors took only one soil sample, divided it in 5 parts, with different degrees of sauration, and procedeed to a comparison between the measures of soil moisture provided by 5 FP sensors and 1 profes-**

[Figure]

**sional sensor. The results of the calibration are depicted in Figure 3. Based on my experience on soil moisture data and measurements provided by professional sensors (TDR in particular) I find this approach particularly risky, in particular because the conditions of soil aggregation (macro pores and other) can strongly change from point to point, also in small scales. I suggest the Authors to elaborate the concept and stress the calibration paragraph, highlighting the need of a more complete (in terms of soil samples) comparison between the sensors. Somehow, I have the feeling that the differences reported in Figure 9 could be attributed to what afore mentioned. Morevoer, I believe the Authors should be more cautious in describing the obtained results in terms of soil moisture also because some errors could have been determined due to the difference in the spatial position between the FP and the professional probes. Indeed the Authors state that "The exact horizontal distance between the FP sensors and the professional probes is unknown...", and that it could be approximately 1 meter. Based on my experience, natural soils (not in laboratories) can present strong differences also in small distances, in particolare concerning the hydraulic conductivity.**

*Response:* We thank the reviewer for the positive feedback, we are pleased that we could attract his/her interest for our study.

We understand the concern about the calibration procedure, thank you for raising this point. In the following we want to clarify our approach and the drawn conclusions.

Indeed, we used one homogenized soil sample, divided it into five parts with different degrees of saturation and used five FP sensors to measure the soil water content. As a reference we did not use a professional sensor, but determined the exact soil water content gravimetrically (p.8, line 174). To evaluate the FP sensors in the field we

used already installed professional sensors, the TDT SPADE sensors and TDR probe ECH2O 5TM. We agree with the reviewer that soil conditions can vary from point to point, certainly at catchment scale and even at much smaller scales. Unfortunately, within the scope of our study it was not possible to conduct the calibration evaluation for multiple soil samples. Although this would have improved the reliability of the FP sensor in terms of absolute soil moisture values, some uncertainty would still have remained (soil conditions can still vary between the removed sample used for calibration and the soil where the sensor is installed in the field). Furthermore, the primary goal of our study was to evaluate the performance of the FP sensor in general, in absolute and relative terms, as not for all purposes absolute soil moisture values may be of highest interest. Of course, if the absolute amount of soil moisture is of interest and not only the relative one, we agree with the reviewer that site-specific calibrations are highly recommended. This recommendation is also included in our conclusions section on page 23, lines 509 to 510.

It is true and we completely agree, that soil conditions in the field can vary strongly even within small distances and consequently can affect the soil moisture measurements. We mention this on page 16, lines 346-349: "In addition, differences in the absolute water content can be caused by local variations in soil texture. Although the FP devices were calibrated for the dominant soil texture type present in the HOAL catchment, the conditions in the laboratory are ideal (e.g. homogeneous soil sample) and less perfect conditions with local variations in soil texture are to be expected in the field." To elaborate on this we suggest to slightly modify this section to: "In addition, differences in the absolute water content can be caused by local variations in soil texture and formed soil aggregates. Although the FP devices were calibrated for the dominant soil texture type present in the HOAL catchment, the conditions in the laboratory are ideal (e.g. homogeneous soil sample) and less perfect conditions with local variations in soil texture and formed aggregates (e.g. even macro pores) are to be expected in the field."

We also agree that the horizontal distance between the professional and the FP sensor can result in the observation of different absolute soil moisture values due to local soil variations. Previously we only implied this fact in the above-mentioned paragraph. For more clarity, we suggest to add the following sentence to the paragraph (page 16, lines 350-351): "This local variation in soil can also differ within the small horizontal distance between the locations of the professional and low-cost sensors and thus can also contribute to the deviation in absolute values between both devices."

We hope that we could satisfactorily address the concerns of the reviewer.